# Dietary Epigenetic Modulators: Unravelling the Still-Controversial Benefits of miRNAs in Nutrition and Disease

**DOI:** 10.3390/nu16010160

**Published:** 2024-01-03

**Authors:** Elisa Martino, Nunzia D’Onofrio, Anna Balestrieri, Antonino Colloca, Camilla Anastasio, Celestino Sardu, Raffaele Marfella, Giuseppe Campanile, Maria Luisa Balestrieri

**Affiliations:** 1Department of Precision Medicine, University of Campania Luigi Vanvitelli, 80138 Naples, Italy; elisa.martino@unicampania.it (E.M.); antonino.colloca@studenti.unicampania.it (A.C.); camilla.anastasio@unicampania.it (C.A.); marialuisa.balestrieri@unicampania.it (M.L.B.); 2Food Safety Department, Istituto Zooprofilattico Sperimentale del Mezzogiorno, 80055 Portici, Italy; anna.balestrieri@izsmportici.it; 3Department of Advanced Clinical and Surgical Sciences, University of Campania Luigi Vanvitelli, 80138 Naples, Italy; celestino.sardu@unicampania.it (C.S.); raffaele.marfella@unicampania.it (R.M.); 4Department of Veterinary Medicine and Animal Production, University of Naples Federico II, 80137 Naples, Italy; giuseppe.campanile@unina.it

**Keywords:** cancer, chronic diseases, epigenetics, microRNA, nutrients

## Abstract

In the context of nutrient-driven epigenetic alterations, food-derived miRNAs can be absorbed into the circulatory system and organs of recipients, especially humans, and potentially contribute to modulating health and diseases. Evidence suggests that food uptake, by carrying exogenous miRNAs (xenomiRNAs), regulates the individual miRNA profile, modifying the redox homeostasis and inflammatory conditions underlying pathological processes, such as type 2 diabetes mellitus, insulin resistance, metabolic syndrome, and cancer. The capacity of diet to control miRNA levels and the comprehension of the unique characteristics of dietary miRNAs in terms of gene expression regulation show important perspectives as a strategy to control disease susceptibility via epigenetic modifications and refine the clinical outcomes. However, the absorption, stability, availability, and epigenetic roles of dietary miRNAs are intriguing and currently the subject of intense debate; additionally, there is restricted knowledge of their physiological and potential side effects. Within this framework, we provided up-to-date and comprehensive knowledge on dietary miRNAs’ potential, discussing the latest advances and controversial issues related to the role of miRNAs in human health and disease as modulators of chronic syndromes.

## 1. Introduction

Among lifestyle factors, diet displays a strong impact on human health. Over decades, unhealthy dietary habits have been responsible for the spread of many severe chronic degenerative diseases such as obesity, type 2 diabetes mellitus (T2DM), cardiovascular diseases, and cancer [1]. Food intake may modify gene expression as well as disease susceptibility through the regulation of several epigenetic modulators [1]. MicroRNAs (miRNAs) play critical roles in gene regulation and biological processes [2,3]. They are transcribed by RNA polymerase II, leading to the generation of a hairpin structure called primary microRNA or pri-miRNA, which is further processed by the RNA endonucleases located in the nucleus (Drosha) and in the cytoplasm (Dicer) to generate a duplex of 22–25 nucleotides [2]. This latter is then divided into two mature sequences associated with the Argonaute protein (AGO), resulting in the formation of the RNA-induced silencing complex (RISC), which can ultimately act as a translation repressor on target genes [3]. Gene expression regulation at intracellular and extracellular levels is improved by mature transcript encapsulation into extracellular vesicles, such as exosomes, secreted by multiple cell types as mediators of cell-to-cell communication [4]. Because of their biocompatibility and biostability, extracellular vesicles are considered regulatory cargos in a wide range of intercellular communication and crosstalk systems [5,6]. MiRNAs have been described in body fluids as blood, serum, plasma, urine, saliva, and breast milk, and changes in their circulating levels are related to a plethora of different chronic syndromes, including obesity, cardiovascular and neurodegenerative diseases, T2DM, and cancer [7,8]. Preserving the human miRNA profile could contribute to the prevention of diseases and the maintenance of good health. MiRNAs have been widely reported in plants, animals, and humans [9]. Plant-derived miRNAs bind to recipient target transcripts with quite perfect complementarity, acting as small interfering RNAs (siRNAs), while animal miRNAs interact with the host mRNA targets with imperfect complementarity, thus inducing their translational repression [10,11]. Given this imperfect complementarity, a single exogenous miRNA (xenomiRNA) is able to recognize multiple target sites and modulate different target genes in the host [10,11]. After ingestion, xenomiRNAs could modulate the gene expression by a cross-kingdom pattern or horizontal transfer of genetic information to the host [10,11]. Bioactive dietary compounds, by affecting directly and indirectly gene activity, have been related to epigenetic changes, such as DNA methylation, histone acetylation, and the modulation of miRNA expression in both physiological and pathological conditions [12,13]. However, the clinical relevance of food-derived xenomiRNAs in human diseases is still undefined. Several studies have demonstrated that the intricate interplay between miRNAs and nutrients could regulate health and chronic diseases, hence pointing to food modification as a pivotal tool in different diseases [14,15]. However, further studies to reveal the precise mode of action of dietary active compounds and how nutrients and bioactive molecules affect miRNA expression are still required. This review aims to provide an update on the role of dietary miRNAs in health and diseases by underlining the benefits, obstacles, and controversies of the relationship between food-derived miRNAs and the pathogenesis of chronic diseases.

## 2. Dietary XenomiRNAs in Health and Disease

XenomiRNAs represent a family of exogenous miRNAs characterized by several dietary sources, animals and vegetables, and are able to integrate into the total miRNA profile of a recipient [16]. Once in the host, these small molecules can be absorbed by the gastrointestinal tract, packaged into vesicles, released into the bloodstream, and delivered to multiple cells and tissues, thus promoting healthy state or affecting the development of chronic diseases, including cancer [17,18,19]. In the following sections, the role of xenomiRNAs from both animals and vegetables as regulators of several chronic conditions will be extensively discussed (Figure 1).

### 2.1. XenomiRNAs from Animal Sources

The following subsection is dedicated to the most studied and characterized xenomiRNAs derived from animal sources and their involvement in chronic syndromes and cancer.

#### 2.1.1. Eggs

Analysis of RNA sequencing revealed the content of several miRNAs, such as gga-mir-2188, gga-mir-30c-5p and gga-mir-92-3p, in the edible parts of chicken eggs, suggesting these noncoding RNAs as interesting tools to take into account for the improvement of egg nutritional value [20]. More recently, Fratantonio et al. described the availability of miRNA-related exosomal vesicles from chicken eggs in mice and humans [21]. Accumulated in the brain, intestine, and lung, miRNA-exosomes regulated spatial learning and memory function in C57BL/6J mice, while egg-derived miRNA levels increased in human peripheral blood following exosomal oral administration (Figure 1) [21].

#### 2.1.2. Meat

The effects of the chronic administration of cooked pork-derived exosomal vesicles were evaluated in mouse model [22]. The upregulated miR-1, miR-133a-3p, miR-206 and miR-99a plasma levels resulted in glucose and insulin metabolism impairment, as well as in lipid droplet accrual in mice liver, supporting the role of pork-derived miRNAs in the development of metabolic disorders (Figure 1) [22]. The bta-miR-154c has been recently characterized by comestible parts of beef. This xenomiRNA is able to contrast the human digestion processes being absorbed from Caco-2, SW480, and SW620 colorectal cancer cells [23]. However, the comprehension of the specific role of miR-154c in colorectal carcinoma requires further investigation [23].

#### 2.1.3. Milk

The classification of milk as an “epi-nutrient”, rich in bioactive compounds and functional molecules, as well as its involvement in counteracting chronic syndromes and cancer, have been recently investigated [24,25,26,27,28]. A total of 678 miRNAs were identified in bovine milk-derived exosomal vesicles [29,30]. These miRNAs are involved in a wide range of cell metabolic pathways, such as miR-181 and miR-155, related to the normal function and differentiation of T and B cells, miR-let-7c, miR-17, miR-92, miR-223 implied in the regulation of immunity and inflammatory cells, and miR-30a, a regulator of autophagy in post-acute myocardial infarction (Figure 2) [31,32,33].

The most abundant miRNA in milk exosomes is miR-148a, able to modulate oral cavity homeostasis as well as to inhibit 5′ AMP-activated protein kinase (AMPK) and phosphatase and tensin homolog (PTEN), both suppressors of mammalian target of rapamycin complex 1 (mTORC1), a pivotal regulator of multiple cell metabolic pathways [31,32,33]. Moreover, milk-derived miR-148a impairs insulin secretion with diabetogenic action through its ability to promote pancreatic β-cell de-differentiation via mTORC1-high/AMPK-low phenotype [33]. It has been reported that miR-148a and miR-21 alter the α-synuclein homeostasis, causing its overexpression and aggregation, with toxic effects on dopaminergic neurons and pancreatic β-cells, thus exerting a possible role in the pathogenesis of Parkinson disease and T2DM [34]. MiRNAs belonging to the miR-148 family affect the immune response, suppressing calcium/calmodulin-dependent protein kinase IIα (CaMKIIα) and the subsequent toll-like receptor (TLR)-mediated expression of major histocompatibility complex II (MHC II) in dendritic cells [31]. Among miRNAs derived from milk exosomes, miR-125a is involved in the modulation of immune response to bacterial and viral aggressions, while the human homolog miR-718, still involved in immune response regulating p53, also regulates vascular endothelial growth factor (VEGF) and insulin growth factor (IGF) pathways [35], and miR-146 exerts a regulatory function in TLR signaling and in the resolution of bacterial infections [36]. As bovine milk, human breast milk-derived miRNAs play a crucial role in modulating development and differentiation of immune system cells and counteracting the onset of metabolic disorders [37,38]. In the first six months of lactation, miR-181a and b, miR-155, miR-125b, and the cluster miR-17-92 actively regulate the T- and B-cell maturation and the tumor necrosis factor α (TNF-α) activation, modulating the immune response of the baby [38], while miR-22-3p counteracts the development of insulin resistance and the onset of T2DM, attenuating the Wnt pathway [37]. Milk-derived miRNAs are also involved in different pivotal metabolic pathways. The presence in milk exosomes of miR-181a-5p has been related to anti-atherogenic effects and reduced vascular inflammation, due to its ability to downregulate nuclear factor kappaB (NF-κB) levels [39]. Another miRNA characterized from milk exosomal vesicles is miR-29, capable of targeting IL-23, a cytokine involved in intestinal damage. To this aim, treatment with milk vesicles containing miR-29 stimulated intestinal stem cell proliferation and gut recovery under several pathological conditions [40], while incubation with miR-31-5p from milk-derived exosomes improved in vitro endothelial function and promoted angiogenesis and diabetic wound healing in vivo [41]. Recent studies reported that oral administration of exosomal vesicles prevented colon shortening, intestinal epithelium disruption, infiltration of inflammatory cells, and tissue fibrosis in a mouse ulcerative colitis model via inhibition of the TLR4-NF-κB signaling pathway and nucleotide-binding oligomerization domain, the NLR family pyrin domain containing 3 (NLRP3) inflammasome activation [30]. Transfection with the milk-derived miR-22 promoted cell proliferation by inhibition of CCAAT/enhancer-binding protein δ (C/EBPδ) expression and promotion of intestinal development in human intestinal epithelial cells [42]. On the other hand, the milk-exosomal miR-148a and miR-30b have been correlated to adipogenic effects, supporting a correlation between milk consumption and obesity incidence [43]. In vitro and in vivo studies reported that milk exosome-derived miRNAs also exert oncogenic or oncosuppressor properties (Figure 2). An association has been reported between cow milk consumption and large B-cell lymphoma development, sustained by miR-148a-3p and miR-155-5p/miR-29b-5p increase via let-7-5p/miR-125b-5p [44]. As oncomir, miR-21 promotes cell growth and anabolism, encouraging cell proliferation and cancer development by activating mTORC1 [45]. MiR-21 and miR-155 have been related to the most advanced stages of breast cancer progression, being involved in the development of tamoxifen resistance, metastasis formation, and worst prognosis [46]. Commercial milk consumption has been correlated to an increased risk of estrogen receptor-positive breast cancer development, due to the content of multiple oncogenic factors, as miR-148a-3p and miR-21-5p [47]. Milk-derived miRNAs have been associated with prostate cancer tumorigenesis promotion. In vitro studies have shown that milk-derived miR-148 increased prostate cancer proliferation by inhibiting cyclin-dependent kinase inhibitor 1B (CDKN1B) and promoted DNA methyltransferase 1 (DNMT1)-dependent epithelial-mesenchymal transition (EMT) [45]. Another significant milk-derived miRNA, miR-125b, increases the development of prostate xenograft cancer targeting proapoptotic genes, as p53, p53-upregulated modulator of apoptosis (PUMA), and BCL2 Antagonist/Killer 1 (BAK1) [48], and regulates several tumorigenic pathways, including NF-κB, p53, phosphatidyl inositol 3-kinase (PI3K)/protein kinase B (AKT)/mTORC1, erb-b2 receptor tyrosine kinase 2 (ERBB2), and Wnt [45]. Epidemiological studies highlighted the association between milk consumption and reduced risk of colorectal cancer [49]. The expression of miR-148a is downregulated in colorectal cancer (CRC), where it exerts a tumor-suppressor activity interfering in NF-κB and signal transducer and activator of transcription 3 (STAT3) pathways and modulating cancer related immune response via inhibition of the programmed death ligand-1 (PD-L1) levels [45]. The in vitro antineoplastic effect of miR-148a-3p overexpression also occurs through mitochondrial impairment, lipid peroxidation, and ferroptosis sustained by the Acyl CoA synthetase long-chain family member 4 (ACSL4)/transferrin receptor (TFRC)/Ferritin axis and direct solute carrier family seven-member 11 (SLC7A11) downregulation in the CRC model [50]. Another buffalo milk-derived miRNA, miR-27b, exerted antineoplastic effects on HCT116 and HT-29 CRC cells by inducing mitochondrial oxidative stress, lysosome accumulation, and apoptotic cell death mediated by endoplasmic reticulum (ER) stress [51].

### 2.2. XenomiRNAs from Vegetable Sources

Several miRNAs characterized from different vegetables and involved in chronic diseases, including cancer, have been evaluated (Figure 1). It has been described the ability of miR-156a, contained in different vegetables, such as cabbage, spinach, and lettuce, to target junctional adhesion molecule A (JAM-A), thus suppressing the development of atherosclerosis in human aortic endothelial cells through the inhibition of monocyte adhesion, occurring under inflammatory stress [52]. Similarly, the miR-167e-5p, characterized by *Moringa oleifera*, exerted a time- and dose-dependent anticancer action in Caco-2 cells acting on the β-catenin pathway [53]. The miR-159, particularly abundant in broccoli, is able to suppress breast tumor development, both in vitro and in vivo models [54,55]. In silico prediction identified two miRNA sequences, bra-miR156g-5p and Myseq-330, in broccoletti *Brassica rapa sylvestris*, as targeting apoptosis-related human genes, although further in vitro studies on pancreatic cancer cells did not support a miR-based modulating role in cancer growth [56].

#### 2.2.1. Rice

An extensive study on miRNAs derived from *Oryza sativa* rice revealed their binding affinity for multiple human genes involved in cardiovascular and neurological diseases and cancer [57]. The *Oryza sativa*-derived osa-miR-172d-5p exerted in vitro beneficial effects in human lung fibroblasts, suppressing transforming growth factor (TGF)-β activated kinase 1 (MAP3K7) binding protein 1 (Tab1) and TGF-β-induced fibrotic gene expression in a bleomycin-induced lung fibrosis model [58]. Analyses on mice and human serum after rice consumption revealed miR-168 circulating levels associated with reduced clearance of plasmatic low-density lipoprotein (LDL) [59,60], while the rice aleurone-derived hvu-miR-168-3p increased the glucose transporter 1 (GLUT1) expression and reduced blood glucose levels by specifically inhibiting the electron transport chain complex I in both in vitro and in vivo models [61].

#### 2.2.2. Ginger

The action of different xenomiRNAs from vegetable sources in the modulation of oncological pathways has been reported. A report on stomach tissues of patients with different gastric conditions, such as gastritis, metaplasia, and cancer, unveiled higher plant-derived miR-168 levels in tissues of patients affected by intestinal metaplasia [62]. In particular, the ginger-derived miR-1078, by regulating leptin, is related to lipopolysaccharide (LPS)-induced interleukin (IL)-6 expression, whereas miR-167a acts as a gut microbiota modulator targeting *Lactobacillus rhamnosus* GG SpaC pilus adhesin [63,64]. Among the different miRNAs contained in the ginger-derived exosomes-like nanoparticles (GELNs), mdo-miR-7267-3p increased the production of IL-22, thus inducing mice colitis [65], while the plant-derived miR-34a, miR-143, and miR-145 act as tumor suppressors in the mouse CRC model [66].

#### 2.2.3. Soybean

In vitro studies showed that the soybean-derived miR-159a exerted an antitumor activity, suppressing Caco-2 proliferation and triggering apoptotic cell death [67], while gma-miR-4995 targets the transcripts of two long noncoding RNAs, metastasis-associated lung adenocarcinoma transcript 1 (MALAT1) and nuclear paraspeckle assembly transcript 1 (NEAT1), strongly expressed during the metastasis formation of several cancer types. The inhibitory action of gma-miR-4995 on MALAT1 and NEAT1 results in the attenuation of cell proliferation and apoptosis induction in CRC cell lines [68]. The soybean exosome-like nanoparticles (ELNs)-derived miR-5781 is able to modulate the inflammatory response by targeting IL-17A [62], while miR-159a might prevent hepatic fibrosis suppressing glycogen synthase kinase-3β (GSK-3β)-mediated NF-κB and TGF-β1 pathways, thus impairing TGF-β1- and platelet-derived growth factor (PDGF)-related hepatic stellate cell activation [69].

#### 2.2.4. Fruits

The ELNs from 11 different edible fruits (blueberry, coconut, grapefruit, Hami melon, kiwifruit, orange, pea, pear, and tomato) contain several miRNAs capable of specifically targeting genes encoding inflammatory mediators [63]. The apple-derived mdm-miR-7121d-h can downregulate the intestinal solute carrier organic anion transporter family member 2B1 (SLCO2B1), thus influencing enteric macromolecules absorption in the human intestine [70].

## 3. Food-Derived Nutrients as miRNA Regulators in Chronic Diseases

Nutrients affect the miRNA profile through the direct or indirect modulation of gene expression. Multiple macro- and micronutrients, such as fatty acids, carbohydrates, vitamins and phytochemicals, are able to regulate miRNA levels [31], thus rendering food epigenetics more intriguing since several pathological patterns and their related pathways are still unravelled. Herein, we report the capability of many phytochemicals, belonging to the macrocategory of polyphenols (resveratrol, curcumin, quercetin, genistein, epigallocatechin gallate), and nonpolyphenols (fatty acids and vitamins), to affect miRNA expression and the involvement of these noncoding RNAs in chronic diseases and cancer (Figure 3).

### 3.1. Inflammatory and Degenerative Diseases

Several bioactive compounds, such as resveratrol, curcumin, polyunsatured fatty acids (PUFA), and quercetin act as antioxidants by modulating the expression of miRNA involved in crucial inflammatory pathways (Figure 3). Resveratrol displayed anti-inflammatory actions by inducing miR-663 and miR Let7A upregulation and a decrease of miR-155 in human monocytes stressed with LPS [71,72,73]. This polyphenolic molecule ameliorated liver fibrosis and reduced hepatocyte apoptosis by inhibiting miR-190a-5p expression, upregulated by profibrogenetic factors such as TGF-β1 and CCl_4_ [74]. The reduction of miR-155 levels to counteract the LPS-induced inflammation has also been reported in macrophages treated with curcumin [75] as well as in mouse macrophages and human microvascular endothelial cells. The treatment with PUFA induced the negative regulation of inflammatory-related miRNAs levels, such as miR-146a, miR-146b, miR-21, miR-125a, and miR-155 [76]. In vitro treatment with apple-derived exosomes displayed anti-inflammatory activity via the promotion of miR-146 expression in human macrophages [77]. Recently, the potential therapeutic role of quercetin, resveratrol, curcumin, and vitamin D against the psoriasis-related inflammatory and proliferative pathogenetic pathways via miR-155, miR-146, and miR-125b regulation has been described [78]. The protective action of quercetin in endometriosis has been reported both in vitro and in vivo. Treatment with quercetin led to miR-145 upregulation, thus negatively regulating the TGF-β1/small mother against the decapentaplegic (SMAD)2/SMAD3 pathway and modulating the pathological process of endometrial fibrosis [79]. Additionally, quercetin downregulated cyclin D1 levels, opposing cell proliferation and inducing apoptosis, by enhancing miR-503-5p, miR-1283, miR-3714, and miR-6867-5p levels [80]. In Alzheimer disease, quercetin exhibited a neuroprotective role in maintaining miRNA homeostasis in neuronal cells by preventing miR-125b, miR-26a, and miR-2218 altered expression [81]. Treatment with resveratrol and selenium nanoparticles reduced metabolic dysfunction and neuroinflammation via increased amyloid-β clearance, sirtuin (SIRT)1 upregulation, and STAT3, IL-1β, and miR-143 downregulation in a rat model of Alzheimer disease [82].

### 3.2. Metabolic Diseases

Many phytochemicals are able to counteract the pathogenesis of metabolic conditions by acting as miRNA regulators (Figure 3). Resveratrol exerted beneficial action on age-related alterations, as senile sarcopenia, by promoting peroxisome proliferator-activated receptor-gamma coactivator 1α (PGC1α) expression and myocyte differentiation by miR-21 and miR-27b upregulation and a decrease of miR-133b, miR-30b, and miR-149 levels [83]. In addition, the resveratrol-induced miR-21 upregulation alleviated cognitive impairment due to insulin resistance and diabetes in mice [84]. The link between saturated fatty acids (SFA) and poor health outcomes and metabolic disorders [85] is already well established; additionally, it is reported that SFA effects on human health could also depend on their influence on miRNAs [86]. Rat myoblast cells treated with palmitic acid (PA) developed insulin resistance and T2DM triggered by miR-29a increase [86]. In addition, PA decreased insulin-induced activation of the PI3K-AKT pathway, enhancing the expression of miR-3180-3p and miR-4632-5p, thus favoring insulin resistance development in HepG2 cells [87]. Stimulation with PA treatment induced mouse cardiomyocyte injuries derived from atrial arrhythmia by targeting miR-27b [88]. High fat and hypercaloric diets can modulate the levels of miRNAs involved in lipid metabolism, cell homeostasis, and fibrogenesis. Evidence demonstrates that fat-rich dietary regimens are associated with downregulation of miR-122 and upregulation of miR-200a, miR-200b, and miR-429 in liver, determining the onset of nonalcoholic fatty liver disease (NAFLD) [89,90]. Similarly, treatment with *Moringa oleifera* prevented liver damage and nonalcoholic steatohepatitis (NASH) progression via SIRT1 upregulation and miR-21a-5p, miR-103-3p, miR-122-5p, and miR-34a-5p downregulation [91]. The role of vitamins in many metabolic processes, as well as in modulating the immune system and disease prevention, has been extensively reported [92]. To this aim, recent reports revealed that treatment with the carotenoid astaxanthin has been able to promote miR-382-5p expression in hepatic stellate cells, thus opposing liver dysfunctional fibrosis [93]. The potential of vitamin D3 and all trans retinoic acid formulations in the prevention of diabetic cardiovascular complications via miR-126 upregulation has been revealed in diabetic mice [94]. Evidence described the ability of quercetin to ameliorate diabetic nephropathy damage by miR-485-5p upregulation and yes-associated protein 1 (YAP1) suppression in human mesangial cells [95] and to attenuate testosterone secretion dysfunction in diabetic rats by reducing ER stress through miR-1306-5p/hydroxysteroid 17-β dehydrogenase 7 (HSD17B7) axis modulation [96]. The combined treatment of catechin epigallocatechin gallate (EGCG) and quercetin prevented insulin resistance by increasing the expression of miR-27a-3p and miR-96-5p, which directly target Forkhead Box O1 (FOXO1), reducing the production of glucose and the transcription of gluconeogenic enzymes [97]. In addition, ECGC showed therapeutic potential in obesity inhibiting white and beige 3T3-L1 and D12 preadipocyte growth by upregulated miR-let-7a and subsequent high-mobility group AT-hook 2 (HMGA2) suppression [98] and inhibiting the MAPK7 pathway by increased miR-143 levels [99]. Clinical data from a cohort of women affected by overweight and insulin resistance revealed that blood orange juice consumption (500 mL per day) for four weeks resulted in upregulated miR-144-3p, miR-424-5p, miR-144-3p, and miR-130b-3p and decreased let-7f-5p and miR-126-3p levels in peripheral blood mononuclear cells, leading to attenuated IL-6 and NF-κB mRNA expression [100].

### 3.3. Cancer

Given the beneficial effects mediated by nutrients on miRNA profile, several studies dissected their use as an oncological approach in different neoplastic contexts (Figure 3).

#### 3.3.1. Digestive System Cancers

Resveratrol exerted anti-inflammatory effects and attenuated colitis-induced tumorigenesis by increasing miR-101b and miR-455 levels [101]. In colorectal cancer, resveratrol decreased the expression levels of characterized oncomiRs, as miR-17, miR-21, miR-25, and miR-92a-2 [102], and exerted a pivotal regulation on many tumor suppressors, such as PTEN and SMAD3, as well as on the oncogenic TGF-β pathway related to progression and metastasis of colorectal cancer by targeting the expression levels of miR-1 and miR-146b-5p [103]. The resveratrol-related modulation of the tumor promoter TGF-β1 was further associated with upregulated miR-663 levels as reported in the SW480 cell line [104]. Genistein exhibited in vitro and in vivo anticancer effect inhibiting miR-95 and its targets AKT and serum and glucocorticoid-regulated kinase 1 (SGK1) on HCT116 cells [105,106] and enhanced miR-1275 levels thus suppressing Eukaryotic Translation Initiation Factor 5A2 (EIF5A2)/PI3K/AKT pathway in hepatocellular carcinoma [107]. In mice colorectal cancer tissues, the walnut-based diet PUFAs have been associated with reduced levels of miR-1903, miR-467c, and miR-3068, along with augmented miR-297a expression levels, resulting in anti-inflammatory, antiangiogenetic, antiproliferative and pro-apoptotic effects [108,109]. In vivo studies showed that vitamin D suppressed colorectal cancer cell proliferation, downregulating histone demethylase 1A (JMJD1A) by increasing miR-627 levels [110,111]. The EGCG treatment was able to suppress miR-483 levels via hypermethylation of its promoter region, modulating the expression of metastatic markers as E-cadherin and vimentin in a mouse model of hepatocellular carcinoma [112], while resveratrol opposed the tumor progression via downregulation of miR-155-5p in gastric cancer cells [113].

#### 3.3.2. Hormone-Dependent Cancers

Resveratrol opposed the overexpressed levels of miR-17, miR-18b, miR-20a, miR-20b, miR-92b, miR-106a, and miR-106b in prostate cancer [114,115], whilst genistein was able to restore in vivo and in vitro miR-574-3p expression levels, generally downregulated in prostate cancer cells [116,117]. Treatment with vitamin D positively regulated the levels of miR-100 and miR-125b in primary prostate cancer tissues, opposing tumor progression [118,119], while in breast cancer T47D and SK-BR-3, cell line treatment with retinoic acid strongly increased miR-10a expression and retinoic acid receptor β (RARβ), whose loss is associated with breast carcinogenesis [120]. Vitamin D was also shown to decrease breast cancer cell capacity to elude natural killer lymphocyte attacks through the reduction of miR-302c and miR-520c expression [121,122]. The effect of curcumin treatment on miR-21 was effective in inhibiting the growth of MCF-7 breast cancer cells [123]. Curcumin treatment also positively regulated miR-181b, miR-34a, miR-16, miR-15a, and miR-146b-5p and down-regulated miR-19a, and miR-19b expression in estrogen receptor-positive primary cells and several breast cancer cell lines with effects on inflammatory cascade, cell cycle progression, survival, and invasiveness [124,125]. The curcumin-dependent miR-146b-5p upregulation suppressed the transactivation of the breast stromal fibroblasts, responsible for the epithelial-to-mesenchymal transition in breast cancer stromal fibroblasts [126]. In vitro studies proved that EGCG counteracted breast cancer progression, attenuating the expression of miR-25 [127], while genistein reduced the expression levels of onco-miR-155, the regulator of several tumor suppressors as PTEN and p27, and impaired cell mobility via p21 activated kinase 2 (PAK2) and miR-23b upregulation in breast cancer cells [128,129]. Resveratrol modulated the oncosuppressor ARH-I expression and limited the “awakening” of ovarian cancer cells, reducing the oncomiR-1305 [130].

#### 3.3.3. Respiratory Tract Cancers

In lung cancer cells, curcumin treatment was able to inhibit cell invasion via upregulation of miR-874, which directly targets matrix metalloproteinase-2 (MMP-2), and miR-98, which suppresses MMP-2 and MMP-9 pathways by targeting LIN28A [131]. In human non-small cell lung carcinoma (NSCLC) A549 cell line, curcumin-induced apoptosis and reduced migration and invasion through miR-206 upregulation and suppression of the PI3K/AKT/mTOR signaling pathway [132]. In addition, treatment with curcumin suppressed tumor progression by increasing miR-192-5p expression and c-Myc reduction in NSCLC cells A427 and A549 [133]. EGCG repressed c-myc expression via upregulation of tumor suppressors miR-let-7a-1 and miR-let-7g in lung cancer cells [134,135]. Quercetin displayed anti-cancer properties by opposing cell survival and enhancing apoptosis in NSCLC cells by increased miR-34a-5p and downregulation of a long noncoding RNA small nucleolar RNA host gene 7 (SNHG7) [136], while in oral squamous cell carcinoma Cal-27 cells quercetin inhibited cell proliferation and metastasis via accumulation of miR-1254 and subsequent downregulation of CD36 [137], and suppressed proliferation and MMP-9 and -2 levels via a miR-16/homeobox A10 (HOXA10) axis increase [138]. In vitro treatment with quercetin promoted apoptosis and repressed metastatic feature in esophageal cancer cells by upregulating miR-1-3p and suppression of Transgelin 2 (TAGNL2) expression [139], while an increase in genistein-induced mir-34a led to production of reactive oxygen species (ROS) and apoptosis in head and neck cancer [140].

#### 3.3.4. Neuroectodermal Cancers

Recent reports described the curcumin-induced miR-222-3p upregulation, thus negatively modulating the expression levels of its target SOX10, causing the inactivation of the SOX10/Notch pathway and limiting proliferation, migration, and invasion of melanoma cells [141]. Similarly, treatment with EGCG upregulated miR-let-7b, thus suppressing growth and development in melanoma cells [142]. Resveratrol induced apoptosis via miR-21 in human glioma cells [143] and genistein inhibited the retinoblastoma cell viability by miR-145 upregulation [144]. The glycoside flavonoid purple sweet potato delphinidin-3-rutin (PSPD3R) suppressed in vitro and in vivo glioma proliferation regulating autophagy by inducing the AKT/cAMP response element-binding protein (CREB)/miR-20b-5p/autophagy related 7 (Atg7) pathway [145].

## 4. Latest Dietary miRNA-Based Animal Models and Clinical Trials

Most of the evidence on the effects of food-borne bioactive compounds on the regulation of human miRNAs in various diseases comes from preliminary in vitro studies, recently validated in animal models. The antidiabetic effects of quercetin and ECGC were proved in mice fed with an enriched-polyphenol diet (0.05% *w*/*w*) ad libitum for 10 weeks [97]. Results revealed the capacity of quercetin and EGCG enrichment to increase miR-27a-3p and miR-96-5p and inhibit gluconeogenesis and FOXO1 pathways, with enhanced effects when combined (Table 1) [97]. BALB/c mice overexpressing or not miR-483-3p were treated for two weeks with ad libitum 0%, 0.1%, or 0.5% EGCG solution, then HepG2 cells were inoculated, and the diet prolonged for two months [112]. Supplementation with EGCG reduced hepatocarcinoma lung metastasis and opposed EMT via miR-483-3p inhibition (Table 1) [112]. Although few clinical trials evaluating the relationship between food-borne miRNAs and human health have been conducted, the number of clinical trials is continuously increasing (www.clinicaltrials.gov) (Table 1). New insights into alterations of skeletal muscle miRNA expression when consuming 1 g/kg/h carbohydrate during the first 3 h of recovery from aerobic exercise (NCT03250234) have been recently described [146]. Authors reported that carbohydrate consumption during an 80 min bout of steady-state treadmill exercise led to increased expression of miR-19b-3p, miR-99a-5p, miR-100-5p, miR-222-3p, miR-324-3p, and miR486-5p immediately following and/or within 3 h from recovery compared with a nutrient-free control, thus resulting in downregulation of breakdown protein gene expression and better muscle recovery (Table 1) [146].

Results from the RESMENA study (NCT01087086) evaluated the effect of the Mediterranean dietary pattern on miRNA levels in white blood cells of 40 patients with metabolic syndrome. Data showed that 8-week hypocaloric diet (30% caloric deficiency) based on Mediterranean diet reduced the expression of miR-155, miR-125, miR-130, miR-132, and miR-422, associated with cancer, atherogenic and adipogenic processes, and other inflammatory conditions (Table 1) [147]. The effects of 60 months with a typical Mediterranean regimen (Med diet) versus a low-fat high complex carbohydrate (LFHCC) diet were correlated to T2DM development and differences in miRNA plasma levels in CORDIOPREV study (NCT00924937) [148]. Following LFHCC consumption, patients with low plasma miR-145 levels showed a higher risk of developing T2DM, as well as the subjects assuming a Med diet with low levels of miR-29a, miR-28-3p, and miR-126 and high miR-150 expression (Table 1) [148]. Current therapeutic options for the treatment of antiphospholipid syndrome showed the efficacy of daily ubiquinol treatment (200 mg for 1 month) in the regulation of a profile of monocyte miRNAs and identified novel and specific miRNA-mRNA regulatory networks associated with atherothrombosis development (NCT02218476) [149]. A completed clinical trial (NCT01634841) provided a systematic investigation of the role of walnuts in preventing or slowing age-related cognitive decline and macular degeneration [150]. Authors screened miRNA profiles in serum samples (before and after intervention) of eight randomly selected participants from the walnut arm, showing 53 miRNAs modulated after one year assuming 15% of daily energy as walnuts, 36 of them being upregulated and 17 downregulated [150]. Compared with participants in the control diet, walnuts consumed for 1 year significantly increased the serum concentration of miR-551 related to the inhibited progression of several cancers [150]. The antineoplastic effects of vitamin D were evaluated in prostate cancer [118]. A cohort of prostate cancer patients was treated with three different vitamin D3 doses (400, 10,000 or 40,000 IU/day) in the time between randomization and prostatectomy (approximately 3–8 weeks). Vitamin D3 assumption determined an increase in tumor suppressors miR-100 and miR-125b in both pathological and healthy prostatic tissue [118]. The HYPODD study correlated the vitamin D supplementation to miR-21 circulating levels in the pathogenesis of cardiometabolic disorders [151]. Hypovitaminosis D correction (50,000 UI/week for 8 weeks and then 50,000 UI/month for 10 months) ameliorated the cardiovascular risk profiles in hypertensive patients without affecting the miR-21 circulating expression (Table 1) [151]. The effects of 4-week time-restricted eating (TRE), a popular form of intermittent fasting, determined downregulation of a miRNA panel (miR-4649-5p, miR-2467-3p, miR-543, miR-301a-3p, miR-3132, miR-19a-5p, miR-495-3p, and miR-4761-3p), which, in turn, could inhibit cell growth pathways while activating cell survival and promoting healthy aging (NCT03590847) [152]. A study conducted on 125 subjects with hypercholesterolemia (ACTRN12619000170123) evaluated the effect of daily assumption of a nutraceutical combination containing 400 mg phytosterols, 100 mg bergamot extract, 20 mg olive extract, and 52 µM vitamin K2 up to 12 weeks of treatment [153]. The evaluated nutraceutical combination did not change the serum lipid profile and inflammation-related miRNAs and biomarkers [153].

**Table 1 nutrients-16-00160-t001:** Dose effect sizes, in vivo models, target miRNAs, and biological relevance to foods, constituents, and dietary regimen or supplementation.

Food, Constituent or Regimen	Dose Effect Size	Model	Target miRNA	Biological Relevance	Refs.
Quercetin	0.05% *w*/*w*	Mouse	miR-27a-3p, miR-96-5p	Antidiabetic	[97]
EGCG	0.05% *w*/*w*	Mouse	miR-27a-3p, miR-96-5p, miR-483	Antidiabetic	[97]
	0%, 0.1%, 0.5% solution ad libitum	Mouse	miR-483	Anticancer properties in hepatocellular carcinoma	[112]
Walnut	30–60 g/d	Human	miR-551	Anticancer properties	[150]
Vitamin D	400, 10,000 or 40,000 IU/day	Human	miR-100, miR-125b	Anticancer properties in prostate cancer	[118]
	50,000 UI/week for 8 weeks, 50,000 UI/month for 10 months	Human	miR-21	Lower cardiovascular risk in hypertensive patients	[151]
Carbohydrates	1 g/kg/h in 3 h after 2 cycle ergometry glycogen depletion	Human	miR-19b-3p, miR-99a-5p, miR-100-5p, miR-222-3p, miR-324-3p, miR-486-5p	Enhanced recovery after exercise	[146]
Ubiquitinol	200 mg/d	Human	20 different miRNAs	Anti-phospholipids syndrome	[149]
Mediterranean diet	8-week hypocaloric diet	Human	miR-155, miR-125, miR-130, miR-132, miR-422	Anti-inflammatory, anticancer, anti-atherogenic	[147]
	low-fat high complex carbohydrate vs. Mediterranean diet for 60 months	Human	miR-145, miR-29a, miR-28-3p, miR-126, miR-150	Diabetic risk	[148]
Nutraceutical combination	400 mg phytosterols, 100 mg bergamot extract, 20 mg olive extract, 52 µM vitamin K2	Human	miR-21, miR-146a miR-126	No significant effects on lipid and inflammatory profile, as on miRNA levels	[153]

## 5. Questions Opening on the Potential Impact of Dietary miRNAs on Health and Disease

Increasing evidence highlight the potential of dietary miRNAs to modulate human pathophysiology, suggesting new dietary-based intervention approaches [154]. However, a preventive strategy based on xenomiRNAs denotes multiple limitations, due to their poor bioavailability, the restricted knowledge of the potential side effects related to high consumption, and the appropriate dietary intake according to their stability due to manufacturing processes and food storage [155]. The bioavailability of food-targeted miRNAs after ingestion is still a controversial issue. In vitro and in vivo evidences reported upregulated miR-29b, miR-200c, miR-21-5p, and miR-30a-5p plasma levels after milk consumption, supporting the availability of these molecules [17,156]. On the other hand, given the base sequence complementarity between miR-21-5p and miR-30a-5p in humans and animals, the precise assessment of xenomiRNA uptaken by diet is quite uncertain [157]. Moreover, the increase in miRNA plasmatic concentration could be ascribed to endogenous responses to other milk-derived compounds [158]. Analyses by real time quantitative PCR (RT-qPCR) and RNA-sequencing on human blood did not provide supporting evidences about the absorption of specific bovine miRNAs after milk consumption [159], as well as results on milk-fed mice highlighted the absence of miRNA absorption and revealed their rapid degradation in intestinal fluids [160]. Analyses by RT-qPCR performed on post-prandial plasma of pigtail macaques reported absent or low-levels of nonspecific amplified miRNAs, thus supporting the scarce intestinal absorption of plant xenomiRNAs [161]. Likewise, the plant-derived miR-168 has been detected in feces and gastrointestinal mucosa while being undetectable in blood, confirming that the potential availability of gastrointestinal tract is not accompanied by systemic absorption [162]. On the other hand, the detection of the plant miR-172 in gastrointestinal tract, serum and feces of mice fed with plant RNA extracts, confirmed the possibility of a dietary-derived miRNA amount absorbed by intestinal tract [163]. Growing evidence reported xenomiRNAs as food contaminants, describing some plant-derived miRNAs as a consequence of experimental contamination and artifacts [19,164]. In addition, the biological and functional role of dietary miRNAs from different sources in human health is still controversial. Different milk-derived miRNAs have been reported to modulate oncogenic and adipogenic pathways, promoting the development of B-cell lymphoma, hormone-dependent breast and prostate cancer [43,44,45,47], as well as the ginger-related miRNA has been associated with colitis development [64] and the pork-derived miRNAs to the development of metabolic disorders in mice [22]. The evaluation of food-derived xenomiRNA effects cannot refrain from considering the co-existence of a great variety of dietary bioactive compounds. In the absence of supplementary intake of dietary exogenous miRNAs, many phytochemicals or animal-based food derivatives may modify the risk of diseases, targeting the reported biological effects [29,165]. Indeed, several food-derived compounds exert their regulatory effects on different diseases either by upregulating or downregulating different signaling pathways, as well as acting on many epigenetic patterns [166]. Dietary constituents are able to affect epigenetic mechanisms, including DNA status modulation, histone methylation, and acetylation, by aiming key epigenetic modulators such as DNA methyltransferases and histone deacetylases, thus resulting in local or global changes in epigenetics and subsequent gene transcription and expression levels [29,165,166]. Attractive strategies from a better knowledge of nutrimiromics are undeniable [167], although further studies are needed to determine the pharmacokinetics and the possible side effects of dietary miRNA-based intervention in human health. Based on current evidence here summarized, it has to take into account both the promising results of food-derived miRNAs and negative effects exerted by the dietary molecules on human health, thus their dual role in chronic diseases still represents an open door in nutrition-based strategies.

## 6. Conclusions

Herein, we provided an up-to-date comprehensive review of food-derived miRNA activities in human health and disease, reporting both beneficial and controversial issues of dietary miRNAs as regulators of chronic conditions. Recent reports aim to elucidate the association between altered miRNA expression and pathological processes contributing to the onset of several chronic diseases. The crucial roles of different miRNAs in the development of T2DM, insulin resistance and obesity [168,169,170,171] have been described, as well as in the progression of other chronic syndromes, such as asthma, allergy, and chronic kidney disease (CKD) [172,173,174,175,176]. This evidence points out nutrient-regulated miRNAs and food-derived miRNAs as intriguing players in food-related health and disease, although the knowledge on miRNA specific role is partially known and still debatable, requiring future studies to broaden knowledge on possible molecular targets and their modulation. In this regard, lifestyle habits, affecting the expression levels of miRNAs, represent a crucial element in the perspective of preventive and personalized medicine. A healthy diet with a personalized choice of nutrients based on individual needs related to age and state of health, i.e., ranging from health to disease or disability/rehabilitative conditions, is essential for maintaining the state of health and to slow down the aging clock. To this end, dietary miRNAs could represent intervention tools in precision nutrition, although with several limitations to overcome, including bioinstability, poor availability, unknown side effects, and multiple molecular targets. Within this framework, future preclinical and clinical studies will benefit from multidisciplinary and translational design taking into consideration some host-related aspects such as microbiome, circadian rhythm and self-susceptibility, and miRNA-derived limitations, as scarce bioavailability and undefined content in body fluids, which undoubtedly add complexity in dissecting the still controversial role of dietary miRNAs in human health and diseases.

## Figures and Tables

**Figure 1 nutrients-16-00160-f001:**
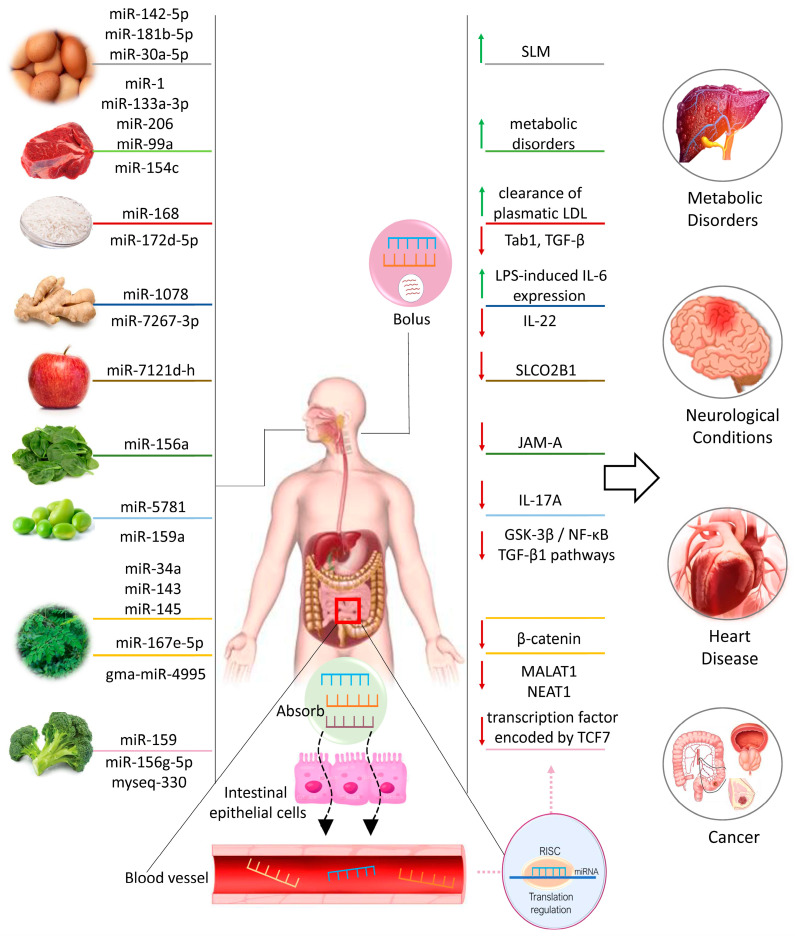
XenomiRNAs from animal and vegetable sources. Exogenous miRNAs from animal and vegetable sources are absorbed by the gastrointestinal tract, packaged into vesicles, and delivered via blood stream to cells and tissues. Here, exogenous miRNAs are able to regulate the development of chronic diseases, such as inflammation, fibrosis, metabolic disorders, neurological conditions, cardiovascular diseases, and cancer, by modulating different cellular pathways. The up arrows stand for upregulation, while the down ones indicate downregulation. GSK-3β, glycogen synthase kinase-3β; IL, interleukin; JAM-A, junctional adhesion molecule A; LDL, low-density lipoprotein; LPS, lipopolysaccharide; MALAT1, metastasis-associated lung adenocarcinoma transcript 1; miR/miRNA, microRNA; NEAT1, nuclear paraspeckle assembly transcript 1; NF-κB, nuclear factor kappaB; RISC, RNA-induced silencing complex; SLCO2B1, Solute Carrier Organic Anion Transporter Family Member 2B1; SLM, spatial learning and memory; Tab1, TGF-β activated kinase 1 (MAP3K7) binding protein 1; TCF7, transcription factor 7; TGF-β1, transforming growth factor-β1.

**Figure 2 nutrients-16-00160-f002:**
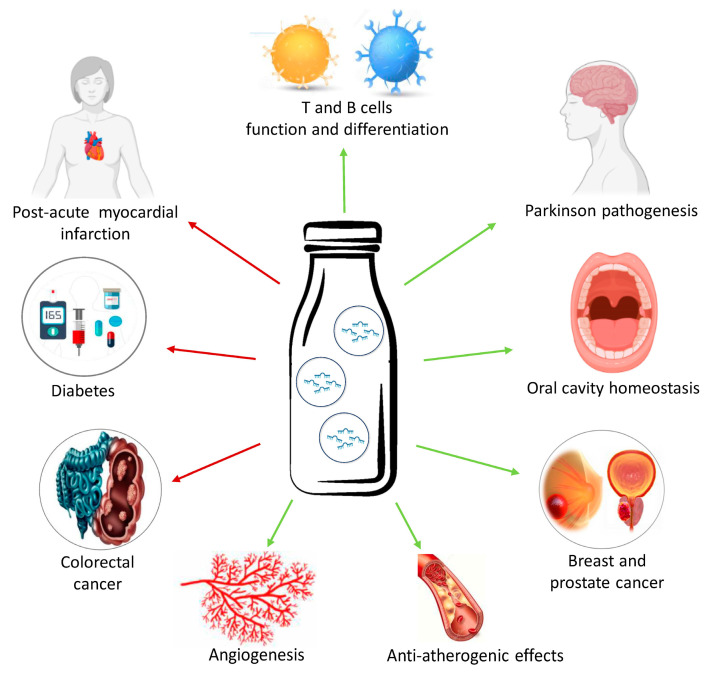
Milk-derived miRNAs and human health. Identified in exosomal vesicles, miRNAs contained in milk are involved in a wide range of metabolic pathways, as the normal function and differentiation of T and B cells, homeostasis of oral cavity, pathogenesis of Parkinson disease, anti-atherogenic action, promotion of angiogenesis, and breast and prostate cancer. MiRNAs contained in milk are also able to counteract diabetes and colorectal cancer development, as well as regulate autophagy in post-acute myocardial infarction. The green arrows stand for promoting activity, while the red ones indicate an opposing role.

**Figure 3 nutrients-16-00160-f003:**
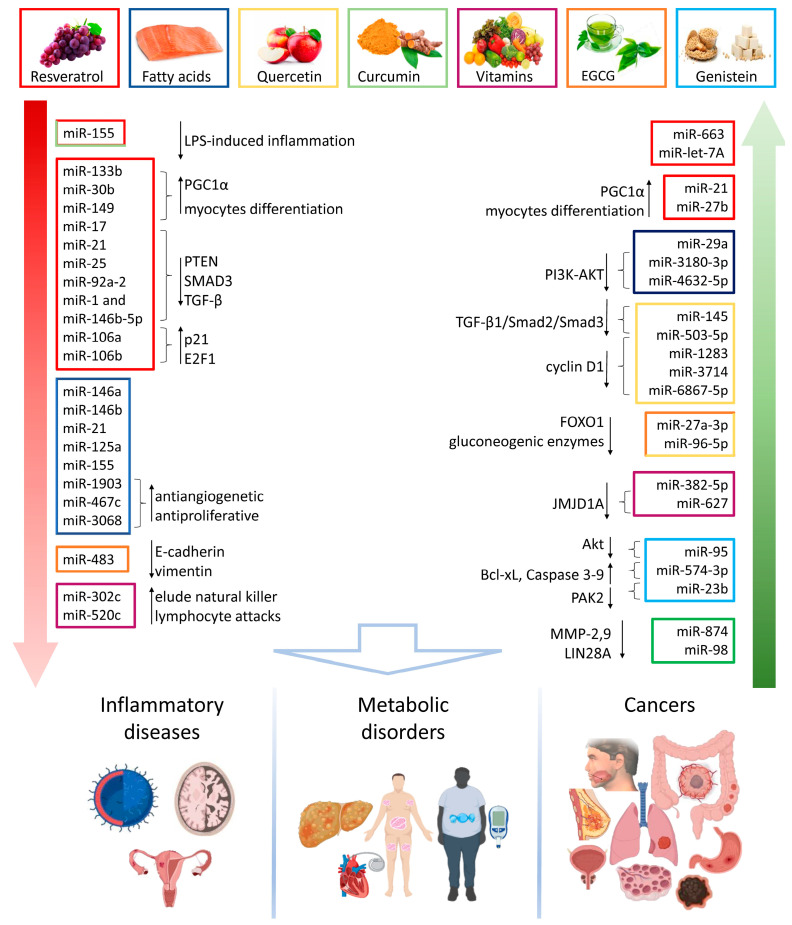
Diet-derived nutrients and miRNA expression. Bioactive compounds, such as curcumin, quercetin, vitamins, fatty acids, EGCG, resveratrol, and genistein, affect human miRNA expression and regulate several pathways, thus modulating the development of pathological chronic states, including inflammatory diseases, metabolic disorders, and cancer. The up arrows stand for upregulation, while the down ones indicate downregulation. AKT, protein kinase B; EGCG, epigallocatechin gallate; FOXO1, Forkhead Box O1; JMJD1A, Jumonji domain-containing 1A; LIN28A, Lin-28 Homolog A; LPS, lipopolysaccharide; miR, microRNA; PAK2, p21 activated kinase 2; MMP, matrix metalloproteinase; PGC1α, peroxisome proliferator-activated receptor gamma coactivator 1α; PI3K, phosphatidyl inositol 3-kinase; PTEN, phosphatase and tensin homolog; SMAD, small mother against decapentaplegic; TGF, transforming growth factor.

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
