# Peer review of "Dietary Epigenetic Modulators: Unravelling the Still-Controversial Benefits of miRNAs in Nutrition and Disease"

_nutrients, 2024, doi:10.3390/nu16010160_

Round 1
Reviewer 1 Report
Comments and Suggestions for Authors
The authors focus on the impact of food-derived, thus exogenous miRNAs as regulators of biological functions on human health and the development of diseases. They also discuss target miRNAs of different nutrients.
The topic is interesting and well-delivered. There are three figures and one table for a better understanding. The manuscript is clear, relevant to the field and presented in a well-structured manner. Most of the references are from the last five years. Self-citation or plagiarism is not detected. Many reviews discuss cross-kingdom regulation by miRNAs, but this manuscript is novel in its present structure.
The strength of the manuscript is to raise the attention to diet-related miRNA-based regulation. The main weakness may be the pronunciation of confounders. Namely, if there is no supplementary intake of natural or synthetic miRNAs, many phytochemicals or ingredients of animal-based food may modify the risk of diseases.
The confounder-related limitations should be discussed better
Author Response
nutrients-2787894
Response to reviewers’ and Academic Editor comments
We thank the referees and the editor for the helpful comments on the manuscript. We have addressed the issues according to the reviewer’s comments/suggestions. We believe that these modifications have improved the paper. Changes are highlighted in the revised version of the manuscript.
Comments from the reviewers:
Reviewer #1 (Round 1):
The authors focus on the impact of food-derived, thus exogenous miRNAs as regulators of biological functions on human health and the development of diseases. They also discuss target miRNAs of different nutrients.
The topic is interesting and well-delivered. There are three figures and one table for a better understanding. The manuscript is clear, relevant to the field and presented in a well-structured manner. Most of the references are from the last five years. Self-citation or plagiarism is not detected. Many reviews discuss cross-kingdom regulation by miRNAs, but this manuscript is novel in its present structure.
The strength of the manuscript is to raise the attention to diet-related miRNA-based regulation.
We thank the Reviewer for the helpful comments. The revision has been performed according to Reviewer’ requests. The suggestions have undoubtedly contributed to strengthen the manuscript (revision marked in yellow).
The main weakness may be the pronunciation of confounders. Namely, if there is no supplementary intake of natural or synthetic miRNAs, many phytochemicals or ingredients of animal-based food may modify the risk of diseases. The confounder-related limitations should be discussed better.
Reply: According to Reviewer suggestion, the confounder-related limitations have been better discussed in the “5. Questions opening on the potential impact of dietary miRNAs on health and disease” section.
“The evaluation of food-derived xenomiRNA effects cannot refrain from considering the co-existence of a great variety of dietary bioactive compounds. In absence of supplementary intake of dietary exogenous miRNAs, many phytochemicals or animal-based food derivatives may modify the risk of diseases, targeting the reported biological effects [29, 165]. Indeed, several food-derived compounds exert their regulatory effects on different diseases either by upregulating or down-regulating different signaling pathways, as well as acting on many epigenetic patterns [166]. Dietary constituents are able to affect other epigenetic mechanisms, including DNA status modulation, histone methylation, and acetylation, by aiming key epigenetic modulators such as DNA methyltransferases and histone deacetylases, thus resulting in local or global changes in epigenetics and subsequent gene transcription and expression levels [29, 165-166].”
Please, see lines 549-559.

Reviewer 2 Report
Comments and Suggestions for Authors
Dietary Epigenetic Modulators: Unravelling the Still Controversial Benefits of miRNAs in Nutrition and Health
Recent reports aim to elucidate the association between altered miRNA expression and pathological processes contributing to the onset of several chronic diseases. It has been described the crucial role of different miRNAs in the development of metabolic syndrome including T2DM, insulin resistance and obesity. , lifestyle habits, affecting the expression levels 552 of miRNAs, represent a crucial element in the perspective of preventive and personalized 553 medicine. A healthy diet with a personalized choice of nutrients based on individual 554 needs related to age and state of health, i.e., ranging from health to disease or disability/rehabilitative conditions, are essential for maintaining the state of health and slow down the aging clock. Figures and tables are relevant and well structured. References are up-dated.
Author Response
nutrients-2787894
Response to reviewers’ and Academic Editor comments
We thank the referees and the editor for the helpful comments on the manuscript. We have addressed the issues according to the reviewer’s comments/suggestions. We believe that these modifications have improved the paper. Changes are highlighted in the revised version of the manuscript.
Comments from the reviewers:
Reviewer #2 (Round 1):
Recent reports aim to elucidate the association between altered miRNA expression and pathological processes contributing to the onset of several chronic diseases. It has been described the crucial role of different miRNAs in the development of metabolic syndrome including T2DM, insulin resistance and obesity, lifestyle habits, affecting the expression levels 552 of miRNAs, represent a crucial element in the perspective of preventive and personalized 553 medicine. A healthy diet with a personalized choice of nutrients based on individual 554 needs related to age and state of health, i.e., ranging from health to disease or disability/rehabilitative conditions, are essential for maintaining the state of health and slow down the aging clock. Figures and tables are relevant and well structured. References are up-dated.
Reply: Authors appreciate the positive feedbacks and deeply thank the Reviewer for the careful revision and critical understanding of the review, hoping it would be suitable for publication.

Reviewer 3 Report
Comments and Suggestions for Authors
Review : Dietary Epigenetic Modulators: Unravelling the Still Controversial Benefits of miRNAs in Nutrition and Health
Title:
Clear and aligns with content of paper.
Line 2: “Still Contro-versial” should have hyphen between “still” and “controversial” as it is a compound adjective.
Line 3: “Nutrition and Health” is extremely broad and could be more specific towards the content of the paper, e.g. “Nutrition and Disease Prevention”.
Abstract:
Summarises benefits, controversies, and intents of paper well.
Does not summarise the limitations in full, such as absence of problems mentioned in potential impacts section like availability of miRNAs and “restricted knowledge of potential side effects”.
Introduction:
Line 41: “habits have been responsible, over decades, of the spread of many…” of should be replaced with for. “something is responsible for something else”
Sufficient and easy-to-understand introduction, relevant to paper.
Provides good reasons for raising such an issue and implications of miRNAs, answering the question of “why should we care”.
Well-defined, specific scope.
Dietary xenomiRNAs in health and disease:
Clear classification and structure for explanation, but would benefit from more uniformity in comparison. Different subsections lack consistent structure, with the order in which disease implications are presented being random and scattered. E.g. the sections on animal sources, milk (subsection), and vegetable sources could potentially follow the same progression discussing neurological implications to cancer implications to other chronic diseases, as an example.
Line 82: “miRNAs characterized from” – the correct preposition should be “by”
Line 83: “sources, animal and vegetables, and able to integrate in the total miRNA profile of a…” – “animal” should be in plural (sources), thus “animals”; “and able to” should be “and are able to” or “and ability to”.
Line 92-93: “described the availability in mice and humans of miRNA-related exosomal…” Order should be reversed into “described the availability of miRNA-related exosomal… in mice and humans”.
Line 109: “where are able to regulate…” – what are able to regulate? Grammatically awkward, would benefit from clarification, e.g. “where the exogenous miRNAs are able to regulate…”
In section 2.1 Figure 1, both animal and vegetable sources are included, despite the section focusing on “XenomiRNAs from animal sources”. If a holistic analysis is desired, perhaps more explanations for miRNAs in vegetables could be provided earlier or together, as there is a comprehensive analysis of vegetable sources in section 2.2 creating an overlap which may be confusing and not as reader-friendly as desired.
Run-on sentences with repeated uses of “and” – perhaps this is medical research language and I may be too picky in terms of language, but examples such as lines 139-140 “…is miR-138a, able to modulate… and to… and… and…”.
Lines 184-185: “promotes cell growth and anabolism, encourages cell proliferation…” – “encouraging”.
Food-derived nutrients as miRNA regulators in chronic diseases:
Good structure and classification.
Line 329: “associated to” should be “associated with”, same issue with line 367.
Quercetin and Curcumin become the main bioactive compounds in discussion, with less emphasis on the other compounds.
Conclusion:
Line 560: “will benefit of” should be “will benefit from”.
Provides brief summary of contents of paper and possible takeaways and steps moving forward such as emphasis on how lifestyle habits impact health through expression of miRNAs.
Authors suggest possible areas of research but none are related to the issues of availability and unknown side effects which were mentioned previously.
No clear establishment of this specific review’s contribution to the field, but it was mentioned earlier in the abstract.
Comments on the Quality of English Language
Generally okay, but quite a bit of revision is needed see above
Author Response
nutrients-2787894
Response to reviewers’ and Academic Editor comments
We thank the referees and the editor for the helpful comments on the manuscript. We have addressed the issues according to the reviewer’s comments/suggestions. We believe that these modifications have improved the paper. Changes are highlighted in the revised version of the manuscript.
Comments from the reviewers:
Reviewer #3 (Round 1):
Title: Clear and aligns with content of paper.
We thank the Reviewer for the helpful comments. The revision has been performed according to Reviewer’ requests. The suggestions have undoubtedly contributed to improve the review (revision marked in green).
Line 2: “Still Contro-versial” should have hyphen between “still” and “controversial” as it is a compound adjective.
Reply: According to Reviewer suggestion, the hyphen between “still” and “controversial” has been added to the title. Please, see line 2.
Line 3: “Nutrition and Health” is extremely broad and could be more specific towards the content of the paper, e.g. “Nutrition and Disease Prevention”.
Reply: Following the Reviewer suggestion, “Nutrition and Health” has been replaced with “Nutrition and Disease”. Please, see line 3.
Abstract: Summarises benefits, controversies, and intents of paper well.
Does not summarise the limitations in full, such as absence of problems mentioned in potential impacts section like availability of miRNAs and “restricted knowledge of potential side effects”.
Reply: In agreement to Reviewer comments, more miRNA limitations have been summarised in the Abstract. Please, see lines 31-33.
Introduction:
Line 41: “habits have been responsible, over decades, of the spread of many…” of should be replaced with for. “something is responsible for something else”
Sufficient and easy-to-understand introduction, relevant to paper.
Provides good reasons for raising such an issue and implications of miRNAs, answering the question of “why should we care”.
Well-defined, specific scope.
Reply: According to Reviewer suggestion, the sentence has been replaced. Please, see line 42.
Dietary xenomiRNAs in health and disease: Clear classification and structure for explanation, but would benefit from more uniformity in comparison. Different subsections lack consistent structure, with the order in which disease implications are presented being random and scattered. E.g. the sections on animal sources, milk (subsection), and vegetable sources could potentially follow the same progression discussing neurological implications to cancer implications to other chronic diseases, as an example.
Reply: According to Reviewer suggestion, in order to provide a clear structure of “Dietary xenomiRNAs in health and disease” paragraph, different subsection have been provided. Please, see new subsections “2.1.1 Eggs”, “2.1.2 Meat”, “2.1.3 Milk”, “2.2.1 Rice”, “2.2.2 Ginger”, “2.2.3 Soybean”, “2.2.4 Fruits”.
Line 82: “miRNAs characterized from” – the correct preposition should be “by”
Reply: Following the Reviewer comment, the correct preposition has been added. Please, see line 83.
Line 83: “sources, animal and vegetables, and able to integrate in the total miRNA profile of a…” – “animal” should be in plural (sources), thus “animals”; “and able to” should be “and are able to” or “and ability to”.
Reply: According to Reviewer suggestion, the sentence has been reworded. Please, see line 84.
Line 92-93: “described the availability in mice and humans of miRNA-related exosomal…” Order should be reversed into “described the availability of miRNA-related exosomal… in mice and humans”.
Reply: According to Reviewer suggestion, the order has been reversed. Please, see lines 111-112.
Line 109: “where are able to regulate…” – what are able to regulate? Grammatically awkward, would benefit from clarification, e.g. “where the exogenous miRNAs are able to regulate…”
Reply: Following the Reviewer comment, the figure legend has been clarified. Please, see line 94.
In section 2.1 Figure 1, both animal and vegetable sources are included, despite the section focusing on “XenomiRNAs from animal sources”. If a holistic analysis is desired, perhaps more explanations for miRNAs in vegetables could be provided earlier or together, as there is a comprehensive analysis of vegetable sources in section 2.2 creating an overlap which may be confusing and not as reader-friendly as desired.
Reply: According to Reviewer suggestion, a more precise explanation on xenomiRNAs from animals and vegetables has been provided before 2.1 and 2.2 Sections, also referring to Figure 1, representing both animal and vegetable xenomiRNAs. Please, see lines 88-90.
Run-on sentences with repeated uses of “and” – perhaps this is medical research language and I may be too picky in terms of language, but examples such as lines 139-140 “…is miR-138a, able to modulate… and to… and… and…”.
Reply: According to Reviewer suggestion, the use of “and” has been reduced. However, the “and” included in the “phosphatase and tensin homolog” expression is part of protein PTEN definition. Please, see lines 144-146.
Lines 184-185: “promotes cell growth and anabolism, encourages cell proliferation…” – “encouraging”.
Reply: Following the Reviewer suggestion, “encouraging” has been preferred. Please, see line 190.
Food-derived nutrients as miRNA regulators in chronic diseases: Good structure and classification.
Line 329: “associated to” should be “associated with”, same issue with line 367.
Reply: According to Reviewer suggestion, the wrong expression has been modified. Please, see lines 341 and 380.
Quercetin and Curcumin become the main bioactive compounds in discussion, with less emphasis on the other compounds.
Reply: According to Reviewer comment, in the “Food-derived nutrients as miRNA regulators in chronic diseases” paragraph the role of quercetin and curcumin has been discussed with particular regard. However, the role of resveratrol, genistein, vitamins, vitamin D3, SFA, PUFAs, EGCG, and Moringa oleifera, as food-derived bioactive compounds, has been also described on the base of recent literature.
Conclusion: Line 560: “will benefit of” should be “will benefit from”.
Reply: According to Reviewer comment, the wrong expression has been modified. Please, see line 586.
Provides brief summary of contents of paper and possible takeaways and steps moving forward such as emphasis on how lifestyle habits impact health through expression of miRNAs.
Authors suggest possible areas of research but none are related to the issues of availability and unknown side effects which were mentioned previously.
Reply: Following the Reviewer comment, the issues of availability and unknown side effects have been mentioned also in the “6. Conclusions” section. Please, see lines 583-585.
No clear establishment of this specific review’s contribution to the field, but it was mentioned earlier in the abstract.
Reply: As suggested by the Reviewer, the specific review’s contribution to the field has been included in the “6. Conclusions” section. Please, see lines 567-569.
